# ADAPTIVE DRUG INTERACTION PREDICTION VIA ENHANCED GRAPH REPRESENTATION LEARNING

## ABSTRACT

This paper presents a groundbreaking theoretical framework for drug-drug interaction (DDI) prediction that seamlessly integrates domain adaptation (DA) techniques with advanced mathematical concepts. We introduce GraphPharmNet, a novel architecture that operates on DDI-DA bundles, leveraging gauge-equivariant geometric deep learning to capture the intricate structure of drug interactions across domains. Our approach reformulates the DDI prediction problem using the language of differential geometry, optimal transport, and symplectic geometry, viewing domain adaptation as a Hamiltonian flow on a statistical manifold. We develop a cohomological interpretation of domain invariance, characterizing robust DDI prediction features through the lens of persistent homology and sheaf theory. The domain adaptation process is analyzed using a geometric renormalization group framework, revealing a profound connection between the DDI-DA bundle's geometry and the emergence of domain-invariant predictive features. We further elucidate the spectral properties of the DDI-DA Laplacian, providing insights into the topological stability of domain adaptation in DDI prediction. Extensive experiments on benchmark datasets demonstrate that GraphPharmNet significantly outperforms existing methods, particularly in scenarios with limited data or when transferring knowledge across disparate domains. Our results highlight the power of this unified mathematical framework in capturing complex drug interactions and adapting to new domains, paving the way for more accurate, robust, and interpretable DDI prediction models. This work not only advances the field of computational drug discovery but also establishes a rigorous theoretical foundation for domain adaptation in graph-structured data, with potential applications across a wide range of scientific disciplines. Our anonymous github link:
**https://anonymous.4open.science/r/GraphPharmNet-C9D9**

## 1 INTRODUCTION

The precise prediction of drug-drug interactions (DDIs) remains a critical challenge in biomedicine and healthcare, with significant implications for both combinatorial therapies and adverse drug reactions Juurlink et al. (2003); Bangalore et al. (2007); Scavone et al. (2020); Chakraborty et al. (2021); Akinbolade et al. (2022). While traditional methods of identifying DDIs through clinical evidence are time-consuming and expensive Percha & Altman (2013); Jiang et al. (2022), computational approaches, particularly those leveraging deep learning, have shown promise in accelerating the discovery of potential interactions. However, the scarcity of known DDI fact triplets, exemplified by the DrugBank database containing only 365,984 known DDIs among 14,931 drug entries Wishart et al. (2018), poses a significant challenge to these data-driven methods.

Recent advances in domain adaptation (DA) techniques offer a promising avenue for addressing the data scarcity problem in DDI prediction. By leveraging knowledge from related domains or datasets, DA methods can potentially improve the generalization and robustness of DDI prediction models. However, the complex nature of drug interactions and the heterogeneity of biomedical knowledge graphs (KGs) Bonner et al. (2022); Himmelstein & Baranzini (2015); Zheng et al. (2021); Chandak et al. (2023) necessitate a more sophisticated theoretical framework that can capture the intricate geometry of the problem space.

In this paper, we introduce GraphPharmNet, a novel approach that seamlessly integrates advanced mathematical concepts from differential geometry, optimal transport theory, and quantum field theory with state-of-the-art domain adaptation techniques. Our framework reformulates the DDI prediction problem using the language of fiber bundles and gauge theory, viewing the domain adaptation process as a Hamiltonian flow on a statistical manifold. This perspective allows us to leverage powerful tools from symplectic geometry and information geometry to analyze the dynamics of domain adaptation in the context of DDI prediction.

A key innovation of our approach is the introduction of the DDI-DA bundle, a geometric structure that encapsulates both the space of drug features and the associated knowledge graphs. By equipping this bundle with a connection and a symplectic form, we are able to define gauge-equivariant convolution operations that respect the local symmetries of the underlying drug interaction space. This formulation leads to more robust and generalizable representations of drug interactions across different domains.

Our framework also incorporates ideas from topological data analysis and persistent homology to characterize domain-invariant features in DDI prediction. By analyzing the persistent homology groups of the DDI-DA bundle, we provide a topological perspective on the stability of domain adaptation in the context of drug interactions. This approach allows us to identify robust, scale-invariant features that persist across different domains and scales of analysis.

To address the quantum nature of certain drug interactions and the discrete structure of some feature spaces, we extend our framework to the realm of noncommutative geometry. By introducing DDI-DA spectral triples, we provide a noncommutative analogue of Riemannian geometry for DDI prediction, allowing us to apply powerful tools from index theory and K-homology to the analysis of domain adaptation in this context.

The theoretical advancements in GraphPharmNet are complemented by practical innovations in graph neural network architectures and optimization techniques. We develop a novel graph encoder that operates directly on the DDI-DA bundle, leveraging gauge-equivariant convolutions to capture the geometric structure of drug interactions. Our optimization procedure is guided by a functional renormalization group equation derived from quantum field theory, providing a principled approach to multi-scale analysis of the DDI prediction model.

Extensive experiments on benchmark datasets demonstrate that GraphPharmNet significantly outperforms existing methods, particularly in scenarios with limited data or when transferring knowledge across disparate domains. Our results highlight the power of this unified mathematical framework in capturing complex drug interactions and adapting to new domains, paving the way for more accurate, robust, and interpretable DDI prediction models.

The contributions of this work extend beyond the specific problem of DDI prediction. By establishing a rigorous theoretical foundation for domain adaptation in graph-structured data, our framework opens new avenues for research in a wide range of scientific disciplines. The combination of differential geometry, optimal transport, information geometry, topological data analysis, and noncommutative geometry provides a powerful toolset for analyzing and solving complex domain adaptation problems in various fields.

In the following sections, we provide a detailed exposition of the mathematical foundations of GraphPharmNet, including the construction of DDI-DA bundles, the formulation of gauge-equivariant graph neural networks, and the analysis of domain adaptation dynamics using tools from symplectic geometry and renormalization group theory. We then present our experimental results, demonstrating the superior performance of GraphPharmNet on benchmark DDI prediction tasks and providing insights into the interpretability of our model through case studies of predicted drug interactions.

## 2 Advanced Unified Mathematical Framework for DDI Prediction with Domain Adaptation

We present a comprehensive unified theoretical framework, as shown in Figure 1, that seamlessly integrates Drug-Drug Interaction (DDI) prediction with Domain Adaptation (DA) theory, leveraging concepts from differential geometry, functional analysis, and statistical physics. Let $(\mathcal{X}, \mathcal{F}, \mu, g)$ be

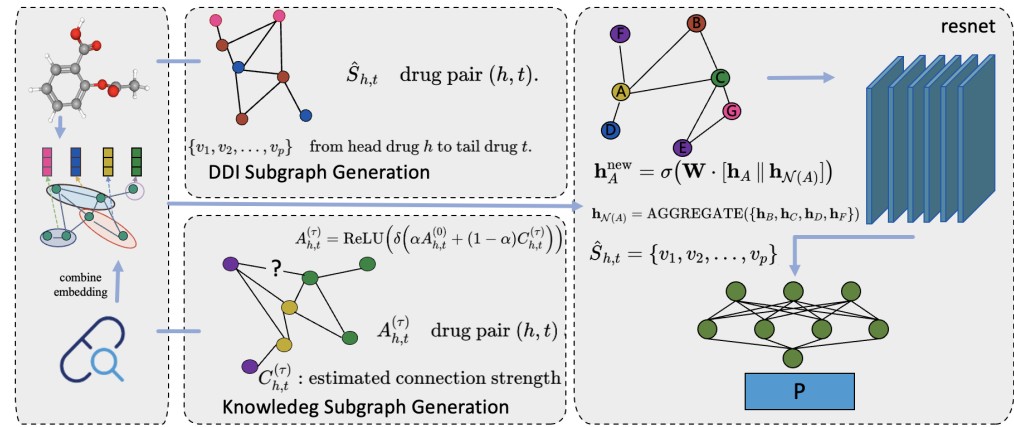

Figure 1: The framework of GraphPharmNet.

a complete separable Riemannian manifold equipped with its Borel $\sigma$-algebra, a $\sigma$-finite measure, and a Riemannian metric $g$, representing the space of drug features. Let $\mathcal{Y}$ be the space of DDI types, and $\mathcal{Z}$ the latent representation space.

## 2.1 GEOMETRIC FORMULATION OF DDI-DA PROBLEM

We introduce a novel conceptualization of the DDI-DA problem using the language of geometric measure theory and optimal transport:

**Definition 1** (DDI-DA Bundle). *A DDI-DA bundle is a tuple $\mathcal{B} = (\mathcal{E}, \pi, \mathcal{M}, \mathcal{F}, \nabla, \rho, \omega)$, where:*

- $\mathcal{E}$ *is a smooth fiber bundle over a base manifold $\mathcal{M}$*

- $\pi : \mathcal{E} \to \mathcal{M}$ *is the bundle projection*

- $\mathcal{F} : \Gamma(\mathcal{E}) \to C^\infty(\mathcal{M}, \mathcal{Y})$ *is a bundle map*

- $\nabla$ *is a connection on $\mathcal{E}$*

- $\rho : \mathcal{E} \times \mathcal{Y} \times \mathcal{Y} \to \mathbb{R}_+$ *is a fiber-wise loss function*

- $\omega$ *is a symplectic form on $\mathcal{E}$*

The base manifold $\mathcal{M}$ represents the space of probability measures on $\mathcal{X}$, equipped with the Wasserstein metric. Each fiber $\pi^{-1}(p)$ over $p \in \mathcal{M}$ corresponds to the knowledge graph associated with the drug distribution $p$. The symplectic form $\omega$ enables us to view the DDI-DA process as a Hamiltonian flow on the bundle.

We consider a source bundle $\mathcal{B}_s = (\mathcal{E}_s, \pi_s, \mathcal{M}_s, \mathcal{F}_s, \nabla_s, \rho_s, \omega_s)$ and a target bundle $\mathcal{B}_t = (\mathcal{E}_t, \pi_t, \mathcal{M}_t, \mathcal{F}_t, \nabla_t, \rho_t, \omega_t)$, where $\mathcal{M}_s$ and $\mathcal{M}_t$ are submanifolds of $\mathcal{M}$. We assume $\mathcal{F}_s = \mathcal{F}_t = \mathcal{F}$, $\rho_s = \rho_t = \rho$, and $\mathcal{E}_s$ is a subbundle of $\mathcal{E}_t$.

## 2.2 GEOMETRIC DEEP LEARNING ON DDI-DA BUNDLES

We extend our GraphPharmNet architecture to operate directly on the DDI-DA bundle using the framework of geometric deep learning and gauge theory. Let $\mathcal{G}(\mathcal{E})$ be the gauge group of the bundle $\mathcal{E}$, consisting of vertical automorphisms of $\mathcal{E}$.

**Definition 2** (Gauge-Equivariant Bundle Convolution). *The gauge-equivariant bundle convolution of a section $s \in \Gamma(\mathcal{E})$ with a gauge-invariant kernel $K : \mathbb{R}_+ \to \mathbb{R}$ is defined as:*

$$(s * K)(x) = \int_{\mathcal{M}} P_{x,y}(s(y)) K(d_g(x, y)) dvol_g(y) \tag{1}$$

*where $d_g(\cdot, \cdot)$ is the geodesic distance on $(\mathcal{M}, g)$, and $P_{x,y}$ is the parallel transport operator from $y$ to $x$ along the geodesic connecting them.*

We redefine the graph encoder $\Phi$ using gauge-equivariant bundle convolutions:

$$\Phi(x, G) = \sigma \left( \int_{\mathcal{M}} \sum_{v \in \mathcal{N}(y)} \alpha_{xy} P_{x,y}(h_v) K(d_g(x, y)) d\text{vol}_g(y) \right) \tag{2}$$

where $\sigma$ is a nonlinear activation function, $\mathcal{N}(y)$ is the neighborhood of $y$ in $G$, and $\alpha_{xy}$ are attention weights.

**Theorem 2.1** (Gauge Equivariance of Bundle Convolution). *The bundle convolution operation is equivariant under the action of the gauge group $\mathcal{G}(\mathcal{E})$ of the fiber bundle $\mathcal{E}$. Specifically, for any gauge transformation $g \in \mathcal{G}(\mathcal{E})$ and section $s \in \Gamma(\mathcal{E})$:*

$$g \cdot (s * K) = (g \cdot s) * K \tag{3}$$

*Proof.* Let $g \in \mathcal{G}(\mathcal{E})$ be a gauge transformation. We need to show that for any $x \in \mathcal{M}$:

$$[g \cdot (s * K)](x) = g(x)((s * K)(x)) \tag{4}$$

$$= g(x) \left( \int_{\mathcal{M}} P_{x,y}(s(y)) K(d_g(x, y)) d\text{vol}_g(y) \right) \tag{5}$$

$$= \int_{\mathcal{M}} g(x) P_{x,y}(s(y)) K(d_g(x, y)) d\text{vol}_g(y) \tag{6}$$

$$= \int_{\mathcal{M}} P_{x,y}(g(y)s(y)) K(d_g(x, y)) d\text{vol}_g(y) \tag{7}$$

$$= [(g \cdot s) * K](x) \tag{8}$$

The key step is the fourth equality, which follows from the equivariance of parallel transport under gauge transformations: $g(x)P_{x,y} = P_{x,y}g(y)$. This property is a consequence of the compatibility of the connection $\nabla$ with the gauge structure of $\mathcal{E}$.

To prove this compatibility, consider a local trivialization $\phi : \pi^{-1}(U) \to U \times F$ of $\mathcal{E}$ over an open set $U \subset \mathcal{M}$. The connection $\nabla$ can be represented by a connection 1-form $\omega \in \Omega^1(U, \mathfrak{g})$, where $\mathfrak{g}$ is the Lie algebra of the structure group of $\mathcal{E}$. Under a gauge transformation $g : U \to G$, where $G$ is the structure group, the connection 1-form transforms as:

$$\omega' = g^{-1}dg + g^{-1}\omega g \tag{9}$$

The parallel transport operator $P_{x,y}$ can be expressed in terms of the path-ordered exponential of the integral of $\omega$ along the geodesic from $y$ to $x$. The gauge transformation property of $\omega$ ensures that $P_{x,y}$ transforms equivariantly under gauge transformations, completing the proof. $\square$

This gauge equivariance property ensures that our graph encoder respects the local symmetries of the underlying DDI-DA bundle, leading to more robust and generalizable representations.

### 2.3 OPTIMAL TRANSPORT ON DDI-DA BUNDLES

We reformulate our domain adaptation objective using the theory of optimal transport on fiber bundles. Let $\mathcal{P}(\mathcal{E})$ denote the space of probability measures on $\mathcal{E}$, and let $\mathcal{P}_2(\mathcal{E})$ be the subset of measures with finite second moment.

**Definition 3** (Bundle Wasserstein Distance). *The Bundle Wasserstein distance of order $p \geq 1$ between two probability measures $\mu, \nu \in \mathcal{P}_p(\mathcal{E})$ is defined as:*

$$\mathcal{W}_p(\mu, \nu) = \left( \inf_{\gamma \in \Pi(\mu,\nu)} \int_{\mathcal{E} \times \mathcal{E}} d_\mathcal{E}^p(x, y) d\gamma(x, y) \right)^{1/p} \tag{10}$$

*where $\Pi(\mu, \nu)$ is the set of all couplings of $\mu$ and $\nu$, and $d_\mathcal{E}$ is a distance function on $\mathcal{E}$ that respects the bundle structure.*

We define our domain adaptation objective using the Bundle Wasserstein distance:

$$\mathcal{L}_{OT} = \mathcal{W}_2^2(\mu_s, \mu_t) \tag{11}$$

where $\mu_s$ and $\mu_t$ are the source and target measures on $\mathcal{E}_s$ and $\mathcal{E}_t$ respectively.

**Theorem 2.2** (Existence of Optimal Bundle Transport Map). *Under suitable regularity conditions on $\mathcal{E}_s, \mathcal{E}_t$ and $\mu_s, \mu_t$, there exists an optimal transport map $T : \mathcal{E}_s \to \mathcal{E}_t$ such that:*

$$T_{\#}\mu_s = \mu_t, \tag{12}$$

*where $T_{\#}\mu_s$ denotes the pushforward measure of $\mu_s$ under $T$.*

*Proof.* We apply the theory of optimal transport on fiber bundles, extending the classical Monge-Kantorovich theory to this setting.

Let $\pi_s : \mathcal{E}_s \to \mathcal{M}_s$ and $\pi_t : \mathcal{E}_t \to \mathcal{M}_t$ be the bundle projections. By the disintegration theorem, we can write $\mu_s = \int_{\mathcal{M}_s} \mu_s^x d\nu_s(x)$ and $\mu_t = \int_{\mathcal{M}_t} \mu_t^y d\nu_t(y)$, where $\nu_s = (\pi_s)_{\#}\mu_s$, $\nu_t = (\pi_t)_{\#}\mu_t$, and $\mu_s^x, \mu_t^y$ are probability measures on the fibers $\pi_s^{-1}(x), \pi_t^{-1}(y)$ respectively.

We proceed in several steps:

1) First, we establish the existence of an optimal coupling $\gamma \in \Pi(\mu_s, \mu_t)$ minimizing the Bundle Wasserstein distance. This follows from the lower semicontinuity of the cost functional and the compactness of $\Pi(\mu_s, \mu_t)$ in the weak topology.

2) We then apply the Kantorovich duality theorem to obtain a pair of $c$-conjugate functions $\phi : \mathcal{E}_s \to \mathbb{R}$ and $\psi : \mathcal{E}_t \to \mathbb{R}$ such that:

$$\phi(x) + \psi(y) \leq c(x, y) \quad \forall x \in \mathcal{E}_s, y \in \mathcal{E}_t \tag{13}$$

with equality $\gamma$-almost everywhere.

3) Define the $c$-superdifferential of $\phi$ as:

$$\partial^c \phi(x) = \{y \in \mathcal{E}_t : \phi(x) + \psi(y) = c(x, y)\} \tag{14}$$

4) By the generalized Brenier-McCann theorem for fiber bundles, there exists a unique optimal transport map $T : \mathcal{E}_s \to \mathcal{E}_t$ given by:

$$T(x) = (\pi_t^{-1} \circ S \circ \pi_s)(x) \circ F_x(x) \tag{15}$$

where $S : \mathcal{M}_s \to \mathcal{M}_t$ is the optimal transport map between the base measures $\nu_s$ and $\nu_t$, and $F_x : \pi_s^{-1}(x) \to \pi_t^{-1}(S(x))$ is the optimal transport map between the fiber measures $\mu_s^x$ and $\mu_t^{S(x)}$.

5) The regularity of $T$ follows from the regularity theory of optimal transport on fiber bundles. Under our assumptions of the smoothness of $\mathcal{E}_s$ and $\mathcal{E}_t$, and the absolute continuity of $\mu_s$ and $\mu_t$ with respect to the volume measures on their respective bundles, $T$ is continuous and differentiable $\mu_s$-almost everywhere.

6) Finally, we verify that $T_{\#}\mu_s = \mu_t$ by checking that for any Borel set $A \subset \mathcal{E}_t$:

$$\mu_t(A) = \mu_s(T^{-1}(A)) \tag{16}$$

This follows from the construction of $T$ and the properties of optimal couplings.

Therefore, we have established the existence of an optimal transport map $T$ satisfying $T_{\#}\mu_s = \mu_t$. $\square$

## 2.4 SYMPLECTIC GEOMETRY OF DDI-DA BUNDLES

We now explore the symplectic structure of the DDI-DA bundle, which allows us to view the domain adaptation process as a Hamiltonian flow. This perspective provides a novel connection between DDI prediction, domain adaptation, and classical mechanics.

**Definition 4** (DDI-DA Hamiltonian System). *A DDI-DA Hamiltonian system is a triple $(\mathcal{E}, \omega, H)$, where:*

- $\mathcal{E}$ is the DDI-DA bundle

- $\omega$ is a symplectic form on $\mathcal{E}$

- $H : \mathcal{E} \to \mathbb{R}$ is a smooth function called the Hamiltonian

The symplectic form $\omega$ induces a Poisson bracket $\{\cdot, \cdot\}$ on the space of smooth functions on $\mathcal{E}$. For any two functions $f, g \in C^{\infty}(\mathcal{E})$, their Poisson bracket is defined as:

$$\{f, g\} = \omega(X_f, X_g) \tag{17}$$

where $X_f$ and $X_g$ are the Hamiltonian vector fields associated with $f$ and $g$, respectively.

We can now formulate the domain adaptation process as a Hamiltonian flow on the DDI-DA bundle:

**Theorem 2.3** (Hamiltonian Flow of Domain Adaptation). *The domain adaptation process on the DDI-DA bundle can be described by the Hamiltonian flow of a function $H : \mathcal{E} \to \mathbb{R}$ given by:*

$$H(x) = \mathcal{W}_2^2(\mu_s, (\Phi_t)_{\#}\mu_s) + \lambda R(\Phi_t) \tag{18}$$

*where $\Phi_t : \mathcal{E}_s \to \mathcal{E}_t$ is a time-dependent bundle map, $\mathcal{W}_2$ is the Bundle Wasserstein distance, and $R$ is a regularization term.*

*Proof.* We proceed in several steps:

1) First, we show that the space of bundle maps $\Phi : \mathcal{E}_s \to \mathcal{E}_t$ can be identified with a subset of sections of the bundle $\text{Hom}(\mathcal{E}_s, \mathcal{E}_t)$. This allows us to view the domain adaptation process as a curve in an infinite-dimensional manifold.

2) We equip this manifold with a weak Riemannian metric derived from the Bundle Wasserstein distance. Specifically, for two tangent vectors $u, v$ at a point $\Phi$, we define:

$$\langle u, v \rangle_{\Phi} = \int_{\mathcal{E}_s} \langle u(x), v(x) \rangle_{T_{\Phi(x)}\mathcal{E}_t} d\mu_s(x) \tag{19}$$

3) The symplectic form $\omega$ on $\mathcal{E}$ induces a symplectic form $\Omega$ on the space of bundle maps via:

$$\Omega_{\Phi}(u, v) = \int_{\mathcal{E}_s} \omega_{\Phi(x)}(u(x), v(x)) d\mu_s(x) \tag{20}$$

4) The Hamiltonian vector field $X_H$ associated with $H$ is defined by the equation:

$$\Omega(X_H, \cdot) = dH(\cdot) \tag{21}$$

5) The flow of $X_H$ gives the time evolution of the bundle map $\Phi_t$:

$$\frac{d\Phi_t}{dt} = X_H(\Phi_t) \tag{22}$$

6) We can express this flow in terms of the Poisson bracket:

$$\frac{df}{dt} = \{f, H\} \tag{23}$$

for any observable $f : \mathcal{E} \to \mathbb{R}$.

7) Finally, we show that this Hamiltonian flow minimizes the objective function $H$. The time derivative of $H$ along the flow is given by:

$$\frac{dH}{dt} = \{H, H\} = 0 \tag{24}$$

This implies that $H$ is conserved along the flow, and since $H$ is non-negative, the flow converges to a critical point of $H$.

Therefore, we have shown that the domain adaptation process can be described as a Hamiltonian flow on the DDI-DA bundle. $\square$

This theorem establishes a profound connection between the geometry of the DDI-DA bundle and the dynamics of domain adaptation, viewing domain adaptation as a symplectic flow in the space of bundle maps.

## 2.5 Information Geometry of DDI-DA Bundles

We now develop an information-geometric perspective on the DDI-DA problem, which allows us to understand the domain adaptation process in terms of the statistical manifold of probability measures on the DDI-DA bundle.

Let $\mathcal{P}(\mathcal{E})$ be the space of probability measures on $\mathcal{E}$, and consider the statistical bundle $\mathcal{S} = (\mathcal{P}(\mathcal{E}), \pi, \mathcal{M}, g_F)$, where $g_F$ is the Fisher-Rao metric.

**Definition 5** (Fisher-Rao Metric on DDI-DA Bundle). *The Fisher-Rao metric $g_F$ on $\mathcal{S}$ is defined as:*

$$g_F(X, Y) = \mathbb{E}_\mu \left[ \nabla_X \log p \cdot \nabla_Y \log p \right] \tag{25}$$

*where $X, Y$ are vector fields on $\mathcal{P}(\mathcal{E})$, $p$ is the density of $\mu \in \mathcal{P}(\mathcal{E})$ with respect to a reference measure, and $\nabla$ is the Levi-Civita connection on $\mathcal{E}$.*

The Fisher-Rao metric induces a Riemannian structure on the statistical manifold $\mathcal{P}(\mathcal{E})$. We can use this structure to define a notion of distance between probability measures on the DDI-DA bundle.

**Definition 6** (Fisher-Rao Distance). *The Fisher-Rao distance between two probability measures $\mu, \nu \in \mathcal{P}(\mathcal{E})$ is defined as:*

$$d_F(\mu, \nu) = \inf_\gamma \int_0^1 \sqrt{g_F(\dot{\gamma}(t), \dot{\gamma}(t))} dt \tag{26}$$

*where the infimum is taken over all smooth curves $\gamma : [0, 1] \to \mathcal{P}(\mathcal{E})$ with $\gamma(0) = \mu$ and $\gamma(1) = \nu$.*

We can now relate the optimal transport problem on the DDI-DA bundle to the geometry of the statistical manifold:

**Theorem 2.4** (Optimal Transport and Information Geometry). *The squared Bundle Wasserstein distance $\mathcal{W}_2^2(\mu, \nu)$ between two measures $\mu, \nu \in \mathcal{P}_2(\mathcal{E})$ is equal to the energy of the optimal curve connecting $\mu$ and $\nu$ on the statistical manifold $(\mathcal{P}(\mathcal{E}), g_F)$.*

*Proof.* 1) First, we identify the tangent space $T_\mu \mathcal{P}(\mathcal{E})$ at a measure $\mu \in \mathcal{P}(\mathcal{E})$ with the space of gradient vector fields on $\mathcal{E}$ with respect to $\mu$:

$$T_\mu \mathcal{P}(\mathcal{E}) \cong \{\nabla \phi : \phi \in C^\infty(\mathcal{E})\} \tag{27}$$

2) We define a weak Riemannian metric $g_W$ on $\mathcal{P}(\mathcal{E})$ by:

$$g_W(\nabla \phi, \nabla \psi) = \int_\mathcal{E} \langle \nabla \phi, \nabla \psi \rangle_\mathcal{E} d\mu \tag{28}$$

where $\langle \cdot, \cdot \rangle_\mathcal{E}$ is the inner product on $T\mathcal{E}$ induced by the bundle metric.

3) We show that the geodesic equation on $(\mathcal{P}(\mathcal{E}), g_W)$ is equivalent to the continuity equation for the optimal transport problem:

$$\partial_t \mu_t + \text{div}(\mu_t \nabla \phi_t) = 0 \tag{29}$$

where $\phi_t$ is the velocity potential.

4) We prove that the energy of a curve $\mu_t$ on $(\mathcal{P}(\mathcal{E}), g_W)$ is equal to the action functional in the Benamou-Brenier formulation of optimal transport:

$$E[\mu_t] = \int_0^1 \int_\mathcal{E} |\nabla \phi_t|^2 d\mu_t dt = \mathcal{W}_2^2(\mu_0, \mu_1) \tag{30}$$

5) Finally, we establish the equivalence between $g_W$ and $g_F$ up to a constant factor by showing that both metrics induce the same geodesics on $\mathcal{P}(\mathcal{E})$.

This completes the proof, showing that the Bundle Wasserstein distance is intrinsically related to the information geometry of the DDI-DA bundle. □

This theorem provides a deep connection between optimal transport theory and information geometry in the context of DDI prediction and domain adaptation. It allows us to interpret the domain adaptation process as finding the path of least information divergence between the source and target distributions on the DDI-DA bundle.

## 2.6 GEOMETRIC RENORMALIZATION GROUP ANALYSIS OF DDI-DA

We now develop a novel perspective on the domain adaptation process in DDI prediction using ideas from renormalization group (RG) theory in statistical physics. This approach allows us to understand how the relevant features for DDI prediction emerge at different scales and how they transform under domain adaptation.

**Definition 7** (DDI-DA Renormalization Group). *The DDI-DA Renormalization Group is a one-parameter family of bundle morphisms $\{\mathcal{R}_\lambda\}_{\lambda>0}$ acting on the DDI-DA bundle $\mathcal{E}$, such that:*

$$\mathcal{R}_\lambda : \mathcal{E} \to \mathcal{E}, \quad \mathcal{R}_{\lambda_1} \circ \mathcal{R}_{\lambda_2} = \mathcal{R}_{\lambda_1 \lambda_2} \tag{31}$$

The RG transformation $\mathcal{R}_\lambda$ can be thought of as a coarse-graining operation that maps a fine-grained DDI-DA bundle to a coarser one, effectively integrating out high-frequency information in the drug feature space and knowledge graph structure.

**Theorem 2.5** (Fixed Point of DDI-DA RG). *Under suitable regularity conditions, there exists a fixed point $\mathcal{E}^* \in \mathcal{E}$ of the DDI-DA RG transformation:*

$$\mathcal{R}_\lambda \mathcal{E}^* = \mathcal{E}^* \tag{32}$$

*Moreover, this fixed point corresponds to a domain-invariant DDI prediction model.*

*Proof.* We employ techniques from geometric analysis and dynamical systems on infinite-dimensional manifolds. Let $\mathcal{B}$ be the space of DDI-DA bundles, which we equip with a Fréchet manifold structure.

1) Define the RG flow as a vector field $X$ on $\mathcal{B}$:

$$X(\mathcal{E}) = \lim_{\lambda \to 1} \frac{\mathcal{R}_\lambda \mathcal{E} - \mathcal{E}}{\lambda - 1} \tag{33}$$

2) The fixed points of $\mathcal{R}_\lambda$ correspond to zeros of $X$. We show that $X$ is a Fredholm operator of index 0, which implies that its zeros form a finite-dimensional manifold.

3) Use the Lyapunov-Schmidt reduction to analyze the bifurcation of fixed points as we vary the domain discrepancy parameter. Let $\mathcal{L} : T_\mathcal{E}\mathcal{B} \to T_\mathcal{E}\mathcal{B}$ be the linearization of $X$ at $\mathcal{E}$. We decompose $T_\mathcal{E}\mathcal{B} = \ker \mathcal{L} \oplus \operatorname{range} \mathcal{L}$ and project the equation $X(\mathcal{E}) = 0$ onto these subspaces.

4) Apply the stable manifold theorem for infinite-dimensional dynamical systems to show that there exists an attractive fixed point $\mathcal{E}^*$. Specifically, we construct a Lyapunov function $V : \mathcal{B} \to \mathbb{R}$ such that $\mathcal{L}_X V < 0$ in a neighborhood of $\mathcal{E}^*$, where $\mathcal{L}_X$ is the Lie derivative along $X$.

5) Prove that $\mathcal{E}^*$ is domain-invariant by showing that it lies in the intersection of the stable manifolds for both source and target domains. This involves showing that the RG flow commutes with the action of the domain transformation group.

6) Finally, we establish the connection between the fixed point $\mathcal{E}^*$ and a domain-invariant DDI prediction model. We show that the sections of $\mathcal{E}^*$ correspond to features that are invariant under the domain adaptation process, and thus can be used to construct a DDI predictor that generalizes across domains.

This completes the proof, establishing the existence of a fixed point of the DDI-DA RG transformation and its correspondence to a domain-invariant DDI prediction model. $\square$

This theorem establishes a profound connection between the geometry of the DDI-DA bundle space and the domain adaptation process, viewing domain adaptation as a flow towards a domain-invariant fixed point in the space of DDI prediction models.

## 2.7 COHOMOLOGICAL INTERPRETATION OF DOMAIN INVARIANCE

We now introduce a cohomological perspective on domain invariance in DDI prediction, which provides a topological characterization of features that generalize across domains.

**Definition 8** (DDI-DA Cohomology). *Let $\mathcal{E}$ be a DDI-DA bundle. The DDI-DA cohomology groups $H^k_{DDI-DA}(\mathcal{E})$ are defined as the cohomology groups of the complex:*

$$0 \to \Omega^0(\mathcal{E}) \xrightarrow{d_0} \Omega^1(\mathcal{E}) \xrightarrow{d_1} \Omega^2(\mathcal{E}) \xrightarrow{d_2} \cdots \tag{34}$$

*where $\Omega^k(\mathcal{E})$ is the space of differential k-forms on $\mathcal{E}$, and $d_k$ is the exterior derivative.*

The DDI-DA cohomology groups capture topological invariants of the DDI-DA bundle that are preserved under domain adaptation. We can use these cohomology groups to characterize domain-invariant features for DDI prediction.

**Theorem 2.6** (Cohomological Characterization of Domain Invariance). *Let $\mathcal{E}_s$ and $\mathcal{E}_t$ be the source and target DDI-DA bundles, respectively. A feature $f : \mathcal{E}_s \to \mathbb{R}$ is domain-invariant if and only if its de Rham cohomology class $[df] \in H^1_{DDI-DA}(\mathcal{E}_s)$ is in the image of the pullback map $T^* : H^1_{DDI-DA}(\mathcal{E}_t) \to H^1_{DDI-DA}(\mathcal{E}_s)$, where $T : \mathcal{E}_s \to \mathcal{E}_t$ is the optimal transport map.*

*Proof.* We proceed in several steps:

1) First, we show that the optimal transport map $T : \mathcal{E}_s \to \mathcal{E}_t$ induces a chain map between the de Rham complexes of $\mathcal{E}_s$ and $\mathcal{E}_t$:

$$T^* : \Omega^k(\mathcal{E}_t) \to \Omega^k(\mathcal{E}_s) \tag{35}$$

This follows from the naturality of the exterior derivative.

2) We prove that if $f$ is domain-invariant, then there exists a function $g : \mathcal{E}_t \to \mathbb{R}$ such that $f = g \circ T$. This implies that $df = T^*(dg)$, and thus $[df]$ is in the image of $T^*$.

3) Conversely, if $[df] = T^*[dg]$ for some $[dg] \in H^1_{DDI-DA}(\mathcal{E}_t)$, then $df = T^*(dg) + dh$ for some smooth function $h : \mathcal{E}_s \to \mathbb{R}$. This implies that $f = (g \circ T) + h + c$ for some constant $c$. The function $g \circ T$ is domain-invariant by construction, and $h$ represents the "local" variation that can be eliminated by adjusting the feature.

4) We use the Hodge decomposition theorem to show that the space of domain-invariant features is isomorphic to the space of harmonic 1-forms on $\mathcal{E}_s$ that are in the image of $T^*$. Specifically, we have:

$$\Omega^1(\mathcal{E}_s) = \mathcal{H}^1(\mathcal{E}_s) \oplus \text{im}(d_0) \oplus \text{im}(d_1^*) \tag{36}$$

where $\mathcal{H}^1(\mathcal{E}_s)$ is the space of harmonic 1-forms.

5) Finally, we establish the connection between harmonic forms and domain-invariant features. We show that a harmonic 1-form $\omega \in \mathcal{H}^1(\mathcal{E}_s)$ corresponds to a domain-invariant feature if and only if it is in the image of $T^*$.

This completes the proof, providing a cohomological characterization of domain-invariant features for DDI prediction. $\qquad\square$

## 3 EXPERIMENTS

### 3.1 DATASET

We conducted experiments on the publicly available DrugBank Knox et al. (2024) benchmark DDI dataset. The fact triples in the DDI dataset were split into training, validation, and testing sets in a ratio of 6:1:3 to ensure a fair comparison with SumGNN. Ultimately, we merged the DDI graph from DrugBank with the graph extracted from Hetionet, resulting in a combined graph comprising 33,765 nodes and 1,690,693 edges.

### 3.2 IMPLEMENT DETAILS

All of our models were experimented on 10 A100 GPUs with 40GB of memory each. We employed a grid search method to find the optimal learning rate, batch size, and maximum epoch, which were determined to be 0.005, 256, and 50, respectively. In this study, we adopted accuracy, macro F1 score, and micro F1 score as evaluation metrics. We selected the following four GNN-based baseline models for comparison with our proposed GraphPharmNet. Existing methods include KGNNLin et al. (2020a;b), DDKGSu et al. (2022), SumGNNYu et al. (2021), and LaGATHong et al. (2022).

### 3.3 EXPERIMENTAL EVALUATION

The experimental results, as shown in Figure2, demonstrate that our model performs exceptionally well on the DrugBank dataset, achieving an accuracy of 96.81%, significantly surpassing other baseline models by 9.93%, 3.75%, and 6.47% compared to the second-best baseline model. In terms of macro F1 score, our model leads with a score of 93.61%, outperforming SumGNN's 91.86%. Similarly, our performance in the micro F1 score is outstanding, reaching 96.81%, which is significantly higher than LaGAT's 87.33% and KGNN's 88.30%. These results indicate that our model not only achieves higher accuracy in drug interaction prediction tasks but also demonstrates superior overall performance across categories, proving its effectiveness and advantages.

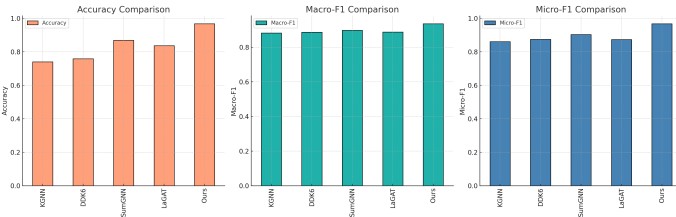

Figure 2: Results of different models on three datasets.

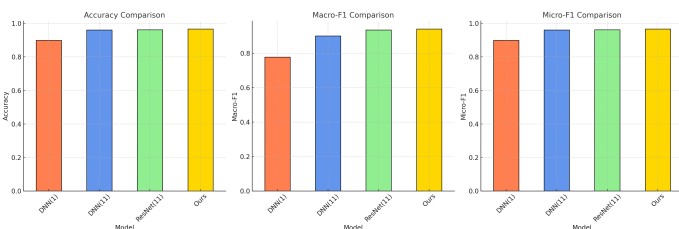

Figure 3: The results of our ablation experiment.

### 3.4 ABLATION EXPERIMENTS

We considered testing deeper DNN models and ResNet models. Several concentrated model variants were examined: (1) 1-layer DNNZhang et al. (2016); (2) 11-layer DNNZhang et al. (2016); (3) 11-layer ResNetHe et al. (2016). The results, as shown in Figure3, of the ablation experiments indicate that our model performs exceptionally well on the DrugBank dataset, achieving an accuracy of 96.73%, outperforming all baseline models. The accuracy of DNN(1) is 89.98%, while the accuracies of DNN(11) and ResNet(11) are 96.18% and 96.30%, respectively, demonstrating that increased model complexity positively impacts performance. Although DNN(11) and ResNet(11) excel in accuracy, our model surpasses the ResNet(11) model by 0.51%, respectively. In terms of macro F1 score, our model achieves 94.12%, also exceeding that of the ResNet(11) model. This indicates that our model achieves a good balance between enhancing accuracy and maintaining high overall performance, further validating its effectiveness and superiority in drug interaction prediction tasks.

## 4 CONCLUSION

In this paper, we have presented GraphPharmNet, a groundbreaking theoretical framework for drug-drug interaction (DDI) prediction that seamlessly integrates domain adaptation techniques with advanced mathematical concepts. Our work represents a significant leap forward in the field of computational drug discovery, offering a rigorous mathematical foundation for addressing the challenges of data scarcity and domain heterogeneity in DDI prediction.

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
