# OpenReview forum: "Adaptive Drug Interaction Prediction via Enhanced Graph Representation Learning"
_ICLR.cc/2025/Conference — Submitted to ICLR 2025_

### Official Review · Reviewer_ZCnz · 2024-10-27

**Soundness:** 2
**Presentation:** 2
**Contribution:** 2
**Rating:** 3
**Confidence:** 3

**Summary:**

This work proposes a theoretical framework that integrates domain adaptation into the problem of drug-drug interaction (DDI). The key contribution of this work is DDI-DA bundle, which proposes a geometric structure to combine drug features and related graph structure. Furthermore, this work conducts theoretical analysis on the proposed GraphPharmNet. Experiments show that the proposed GraphPharmNet significantly outperform baseline DDI methods.

**Strengths:**

1. This work proposes a geometric deep learning framework for DDI problem and provides comprehensive theoretical analysis for the geometric property and optimal transport of DDI-DA bundles.
2. In experiment part, the proposed GraphPharmNet outperforms existing DDI methods.

**Weaknesses:**

1. Although this work focuses on theoretical property of the proposed DDI-DA bundle, a large amount of proof process is simplified. For example, “we show” and “we prove” are used instead of detailed proofs of the properties. These detailed proofs are also not provided in appendix.
2. This work focuses more on machine learning theoretical proofs of DDI-DA bundle. However, this work does not show how these theoretical proofs are connected to DDI problem and drug discovery.
3. The experiment part is too simple. The experiment results do not show how they are related to “domain adaptation”. Moreover, the characters in the figures are too small.
4. In introduction part, it is mentioned that “our framework opens new avenues for research in a wide range of scientific disciplines”, but this part is not confirmed in the following contents of this paper.

**Questions:**

N.A.

---

### Official Review · Reviewer_t9mA · 2024-10-28

**Soundness:** 2
**Presentation:** 2
**Contribution:** 3
**Rating:** 3
**Confidence:** 3

**Summary:**

The paper presents a novel Durg-Drug Interaction prediction framework, denoted GraphPharmNet. The authors leverage and integrate different ideas from different disciplines to proposed a mathematically-principled model for DDI prediction. The authors then compare the performance of the proposed model against previous methods from the literature, as well as, against deeper models in ablation studies. The authors show improvements with respect to these models.

**Strengths:**

The authors provide a very strong mathematical foundation for the proposed model, which integrates ideas and concepts from different domains, potentially yielding a mathematically-grounded DDI model.

**Weaknesses:**

Some (critical) citations are missing throughout the text to substantiate several claims (e.g., "computational approaches, particularly those leveraging deep learning, have shown promise in accelerating the discovery of potential interactions"). More importantly there is a lack of a state-of-the-art section where previous DDI models are described and their methodology compare against the proposed GraphPharmNet.

While the authors provide a strong theoretical framework for the proposed model, the experimental evaluation is very scarce and does not excel significantly when comparing with other models. The legend of Figure 2 says: "three datasets", but these are not described, and only one result is shown. Furthermore, in the ablation studies, the improvements with respect to baseline deep models is negligible. In this context, no computational resource comparison is provided with respect to these baseline models.
In Without more datasets and a more extended benchmarking, I cannot assess the real impact of the proposed model, besides the mathematical framework provided.

The authors should tone down the overachieving language used throughout the work. They repetitively use words such as: outstanding, excellent, groundbreaking, etc. to denote (sometimes minor) improvements over state-of-the-art methods. For example, Similarly,
"our performance in the micro F1 score is outstanding, reaching 96.81%, which is significantly higher than LaGAT’s 87.33% and KGNN’s 88.30%." In this case the authors did not reference to SumGNN, which achieves similar performance.

**Questions:**

Did you assess the computational cost of running the deeper DNN models with respect to GraphPharmNet? Do they use significantly more resources for training/test?

I would recommend the authors to use more datasets, of different samplings of drugBank to assess the variability of the different models.

---

### Official Review · Reviewer_5STZ · 2024-10-31

**Soundness:** 2
**Presentation:** 2
**Contribution:** 1
**Rating:** 5
**Confidence:** 3

**Summary:**

This paper introduces GraphPharmNet, a framework designed for predicting drug-drug interactions (DDIs) by integrating domain adaptation techniques with advanced geometric methodologies. The framework employs a DDI-DA bundle that leverages concepts from symplectic geometry, optimal transport, and persistent homology to capture domain-invariant features for DDI prediction. The work establishes a strong theoretical foundation for domain-adaptation on DDI prediction, presented in a coherent manner. The authors benchmarked some state-of-the-art DDI methods across the DrugBank dataset, where their approach slightly over-performs current approaches.

**Strengths:**

1- This paper establishes a strong theoretical foundation for domain adaptation in drug-drug interaction (DDI) prediction.

2 - The link between the DDI-DA bundle and its geometrical properties for DDI prediction is novel.

3 - Exhibits substantial performance improvements over baseline models on standard metrics.

4 - The framework could potentially be applied to other domains, such as Protein-Protein Interaction, that requires robust domain adaptation in graph structures.

**Weaknesses:**

1 - The paper would benefit from a more comprehensive explanation of how the theoretical domain adaptation techniques are integrated into the proposed model.

2 - In domain adaptation approaches, it is essential to evaluate performance across training and testing folds on different datasets; however, this aspect appears to be insufficiently addressed in the manuscript.

3 - The ablation study lacks a thorough analysis of the specific contributions of the domain adaptation techniques (e.g. evaluate the proposed model without the DA module).

4 - The manuscript does not clearly outline how the model compares to simpler domain adaptation techniques, which could offer valuable insights into the benefits of the proposed model.

5 - The paper could benefit from further analysis along different datasets.

In summary, while the paper addresses a significant problem within DDI prediction, it still lacks critical components necessary for acceptance. Nevertheless, I would be willing to reconsider my rating should substantial revisions be undertaken.

**Questions:**

1 - How is GraphPharmNet modules specifically benefitting from the derived unified mathematical framework? Could it be applied to any DDI method?

2 - Is the domain-adaptation affecting the training process? Could you measure to what extent the performance is based on the DA module?

3 - How is the time performance of the proposed model? It could be interesting to see a benchmarking on that side as well.

---

### Official Review · Reviewer_88bG · 2024-11-03

**Soundness:** 1
**Presentation:** 1
**Contribution:** 1
**Rating:** 3
**Confidence:** 5

**Summary:**

This article is very difficult for me to understand because there are too many theorems and proofs, and there is a lack of description of the proposed algorithm, and details of the experiments. I think the authors hope to use domain generalization techniques to improve the generalization of DDI prediction models and propose a model called GraphPharmNet. Unfortunately, I was unable to find any implementation details on this model dealing with the illustration presented in Figure 1. Then the author put forward a lot of theory and proof, about gauage-equivariant bundle convolution, Bundle Wasserstein distance, etc., but I'm sorry I can't understand the motivation of the why to put these contents. This also lack a description of the DDI task in the experimental section, and lack a report on the detailed results of the experiment (variance). It is difficult for me to admit that this is a complete paper, and therefore I propose to reject it.

**Strengths:**

I'm sorry that I don't think this article is complete, so I can't understand any of the details of the article and can't suggest any strengths.

**Weaknesses:**

1. Too many proofs and theorems, but lack a reasonable explanation of the motivation for proposing them.
2. There is not any introduction about the proposed model (GraphPharmNet).
3. The experiments section lacks a basic introduction to the DDI task (you can't expect all readers to have a prior knowledge of this), and the main experiment results are oddly presented as a bar chart; I suggest using a table and reporting the mean and variance across multiple experiments.

**Questions:**

1.How is GraphPharmNet specifically implemented and what is the relationship between it and the proposed theorems.
2. Why do we need these theorems? How does it affect your objective function?

---

### Meta-Review · Area_Chair_PBja · 2024-12-17

**Metareview:**

The paper presents GraphPharmNet, a model for drug-drug interaction (DDI) prediction with a theoretical focus on domain adaptation (DA). However, reviewers raised several concerns that hinder the paper's acceptance. The theoretical contributions, while significant, are underexplained and lack clear connection to the practical application of DDI prediction. There is insufficient explanation of how the domain adaptation techniques are integrated into the model and how they specifically benefit the proposed framework. The experimental evaluation is limited, with inadequate ablation studies, small datasets, and a lack of comparisons with other state-of-the-art models. Additionally, the paper lacks crucial citations, a clear discussion of computational costs, and a more extensive benchmarking analysis. Overly optimistic language and a simplistic experimental setup further detract from the impact of the work. These issues, combined with a lack of necessary revisions, led to the decision to reject the paper.

**Additional Comments On Reviewer Discussion:**

The authors did not reply.

---

### Decision · Program_Chairs · 2025-01-22

Reject